# Positive Impact of Home ERT for Mucopolysaccharidoses and Pompe Disease: The Lesson Learnt from the COVID-19 Pandemic

**DOI:** 10.3390/healthcare11081176

**Published:** 2023-04-19

**Authors:** Agata Fiumara, Giuseppina Lanzafame, Annamaria Sapuppo, Alessia Arena, Lara Cirnigliaro, Rita Barone

**Affiliations:** 1Regional Referral Centre for Inborn Errors Metabolism, Pediatric Clinic, Department of Clinical and Experimental Medicine, University of Catania, 95123 Catania, Italy; giusylanz@virgilio.it (G.L.); annamaria.sapuppo@unict.it (A.S.); alessia.arena@gmail.com (A.A.); 2Child Neurology and Psichiatry Section, Department of Clinical and Experimental Medicine, University of Catania, 95123 Catania, Italy; lara.cirnigliaro@phd.unict.it

**Keywords:** lysosomal storage disorders, enzyme replacement therapy, home infusion, Pompe disease, Mucopolysaccharidoses, COVID-19

## Abstract

Objective: Patients with Lysosomal disorders (LSDs) are treated with regular infusions of enzyme replacement therapy (ERT). During the COVID-19 pandemic, home treatment was permitted. This study aimed at monitoring the patients’ compliance with home therapy and its effects on physical, psychological, and relational issues. Moreover, we also tested the possible impact of home therapy on familial relationships and contacts with the referral hospital. Materials and Methods: Thirteen patients with Pompe disease (N = 8) and MPS (N = 5) were tested through an online questionnaire designed to assess their level of appreciation and satisfaction with home therapy and their feelings about the referral centre and psychological support provided. Results: Most of the patients (84%) stressed the positive impact of home therapy. All patients described a significant reduction in stressful conditions associated with the need to attend the hospital every week or two. Conclusions: Home ERT leads to a clear improvement in “daily life skills”, as represented in our by sample by positive feelings, better emotional self-control, and an increased ability to understand the feelings of relatives. Our data underline the paramount positive effect home ERT has on both patients and their families.

## 1. Introduction

Lysosomal storage disorders (LSDs) are genetic metabolic errors leading to progressive multisystem involvement. These diseases are due to genetically determined defects of lysosomal enzymes (or of their activators or transporters) that cause the progressive storage of undegraded substrates in different tissues and organs [1,2]. Affected patients show a broad spectrum of clinical signs linked to the main sites of expression of the specific enzyme and the different amounts of residual enzyme activity.

Pompe disease, glycogenosis type II, is due to acid maltase deficiency, with glycogen accumulation in lysosomes mainly involving the skeletal and cardiac muscles. Early-onset-type infantile-onset Pompe disease (IOPD) manifests in the first few weeks or months of life with weakness, hypotonia, developmental delay, macroglossia, and consequent feeding difficulties and failure to thrive. Late-onset forms are characterized by a slowly progressive hypotonia clinically resembling that of other progressive myopathies (limb girdle dystrophy or Duchenne/Becker dystrophies) [3].

In Mucopolysaccharidoses (MPSs), undegraded complex sugar molecules (glycosaminoglycans, GAGs) are stored in connective and other body tissues, mainly the skin, cartilage, liver, and spleen. Different types are characterized by prevalent skeletal or central nervous system involvement [1].

Although rare as single diseases, the overall incidence of LSDs can reach 1/5000 newborns [4]; thus, newborn screening is under consideration in many countries [5,6,7]. At present, no definitive cure is possible. However, thanks to a growing interest in rare diseases over the past years, enzyme replacement therapy (ERT) and other possible therapeutic approaches (e.g., chaperone therapy and gene therapy) have been considered [8].

Since the late 1990s, ERT has become available for some of these diseases; thus, patients with different treatable lysosomal storage disorders are infused with the missing enzyme on a weekly or bi-weekly basis. This treatment is regularly offered by local referral centres.

Since its unexpected outbreak in March 2020, the COVID-19 pandemic has subverted our daily routines, lifestyles, work dynamics, education, and social interactions. A huge impact has been observed on patients with rare lysosomal diseases who require regularly scheduled ERT.

Hosting referral centres had to cope with new restrictions and follow accurate rules to admit patients; conversely, the fear of infection led to a sudden refusal in attending the hospital for ERT as regularly as before. In a recent study [9], we observed that 60% of these patients declined to visit the hospital despite being aware of the precautions and aseptic procedures adopted by the personnel. In a structured interview, the patients all stated that they were highly scared to become infected and die, although they were well aware of the risks linked to ERT interruption in terms of disease progression.

In the same study, we observed a profound negative self-assessment, with reports of a sad feeling with respect to relationships with others and the future, which resignedly felt uncertain and unsafe. All interviewed subjects, both the patients and their relatives and caregivers, stated a need for psychological support.

In March 2020, AIFA (Agenzia Italiana del Farmaco), the Italian Medicines Agency approved a special license for home therapy during the pandemic (DET 341/2020), allowing for ERT administration at home for treatments originally dispensed only at the hospital. Thanks to this statement, all patients regularly treated in our centre were transferred to home therapy with the help of dedicated sanitary teams. Although the Italian health service system does not support ERT at home, home ERT was realized with the help of experienced teams (tutors and HNPs) sponsored by two pharmaceutical-producing companies, Sanofi and Biomarin.

After one year and again after two years, due to the continuing alert for COVID-19, we decided to monitor the patients’ compliance to home therapy and its effects on physical, psychological, and relational aspects. Moreover, we also assessed the caregivers’ feelings and how they felt about the relationship with the referring hospital.

## 2. Materials and Methods

Several patients with different forms of LSD are followed at our Regional Referral Centre for Metabolic Diseases. Informed consent is regularly obtained at the beginning of treatment and each new year when a new medical record is opened. Patients with Gaucher disease and Fabry disease were not enrolled in the study because they were already receiving home ERT before the pandemic emergency. All enrolled subjects provided their informed consent for inclusion before they participated in the present survey. The study was conducted in accordance with the Declaration of Helsinki, and the protocol was approved by the Ethic Committee of AOU Rodolico San Marco, Catania (OBS17128). Our sample includes thirteen patients: eight with Pompe disease (PD) and five with different forms of Mucopolysaccharidoses (MPS) comprising three MPS VI and two MPS IV patients who agreed to be enrolled in the study. They were seven females and six males aged 5 to 54 years. Eleven patients personally responded to our interview, while the main caregiving relatives of the remaining two patients were interviewed due to the patients’ age and/or physical conditions. The interview was conducted after one year of home therapy and again at the end of the second year. An online questionnaire (see the Appendix A) was developed based on 17 questions. Fourteen questions were of the “closed” type, with multiple-choice responses or assessment scales to assess appreciation and satisfaction with home therapy, the relationship with the referral centre, and the psychological support provided. We also paid attention to the changes felt and the beneficial effects, emotions, and motivations of the responses provided. Three other questions were “open” and were related to future planning and perspectives and suggestions for improving the behaviour of the Referral Centre and its operators.

## 3. Results

The home therapy staff realized a monitoring system, updating us regularly about adverse events (AEs) and of any change to the ERT infusion calendar to obtain our approval. No significant AEs were registered for any of our patients, and no patient contracted a COVID-19 infection during the time of investigation.

The number of missed infusions was very low and was limited to the transition period from the hospital to home ERT at the beginning of the pandemic emergency.

Most of the patients (84%) stressed the positive impact of home therapy.

All felt positively about changes in routine infusions, although few of patients (8%) denied significant changes or were uncertain about the evaluation (8%).

In particular, 46% stated that they felt a positive psychological effect from these changes, and another 45% reported a better feeling concerning their physical, psychological, and relational state. Of the interviewed patients, 9% described an improvement in both the psychological field and in their familial relationships No other changes described in the questionnaire were reported (Figure 1).

Changes related to home therapy were judged satisfactory by 58% of our sample and fully satisfactory by the remaining 42%. All participants described a significant reduction in stressful conditions associated with the need to visit the hospital every week or two. Moreover, they underlined the reduced risk of infection and the comfortable environment experienced while being treated at home. All these aspects had positive physical and psychological impacts on the whole family.

Concerning the Centre’s role in realizing this opportunity, 62% of patients scored the Centre as optimal and 23% scored it as good in that the staff was depicted as “professional, honest and efficient, paying attention to the patient needs”; nevertheless, 15% of the patients complained of bureaucratic delays.

All patients appreciated the psychological help and guidance offered both pre- and post-transition to their homes. Of the participants, 46% declared they were fully satisfied, 39% were quite satisfied, and 15% were a bit satisfied (Figure 2A). Changes deriving from the psychological support were judged as having a positive impact on psychological health by 67% of the sample, and as positive both psychologically and physically by 17%, while 8% stated that they had observed positive changes in family relations, and the remaining 8% experienced a significant improvement in all fields (physical, psychological, and family relations), as reported in Figure 2B.

In 2022, the risk of a new lockdown linked to a further increase in positive COVID-19 cases evoked different and variegated emotions: 22% complained of anxiety, 22% complained of fear, and 22% complained of distress, and 11% reported resignation. Surprisingly, 11% of patients stated that they were ready to face another lockdown with optimism, 6% with surprise, and 6% as a challenge (Figure 3).

Concerning projects for the future, all patients stated that they were ready to face any further emergency, given that they were used to living “day by day” because of their rare disease.

Another question regarded possible advice to improve the Centre efficiency. Although there was a wide appreciation of professional, human, and caring characteristics attributed to the personnel, all patients stressed the need for a closer contact and a one-to-one relationship, if possible, with the attending physician, as well as more privacy during the infusions and psychologist sessions.

When retested by phone at the end of the second year, every patient underlined the usefulness and the importance of being treated at home. Most of the interviewed patients reported increased physical and mental strength and a global improvement in familial relationships. All of them reported being fully satisfied with home treatment thanks to the well-organized home assistance.

The patients also stressed the usefulness and importance of the online psychological support, which offered human contact in the form of a specialist that listened to them and monitored their health state and was regarded as a strong link to the referral Centre.

## 4. Discussion

Patients with rare lysosomal disorders require special attention in terms of specific, costly drugs, control visits, and exams, and mainly as individuals. Most of the time, the need to comply with therapeutic schedules subverts their lives and those of their caregivers and relatives. Another important concern is the increased cost in terms of expenses for travel and the loss of working or school days both for the patients and their caregivers. Moreover, the routine working days of “metabolic” physicians, who are often overloaded by many organizational issues and few experienced co-workers, impair the realization of the ideal assistance setting and care. Indeed, all patients stressed the need for a closer contact with the caring physician and asked for more privacy during infusions and psychologist sessions.

Thus, patients with lysosomal diseases complained of many unmet needs. In a recent study, Gufon N et al. [10] reported on patients with different types of Mucopolysaccharidoses, stressing that both patients and caregivers expect to have better support by talking with doctors, planning the transition from paediatric to adult medicine, and programming the schedule of infusions according to their commitments. The same authors stress the importance of paying attention to the psychosocial status of the patients and their caregivers [10].

Since the beginning of the COVID-19 pandemic, patients with rare diseases have been required to face one more challenge.

Kristal et al. in Israel [11], who evaluated the effects of the COVID-19 pandemic on patients with lysosomal diseases, reported that almost 25% of ERT infusions were missed. Moreover, they described some mood changes associated with different degrees of motor and cognitive deterioration during the lockdown. This was a common issue described by many centres worldwide, and different percentages of missed infusions were reported, ranging from 20 to 77% [12,13]

In Italy, thanks to AIFA regulations, most patients with LSD benefited from home ERT. Very few infusions were missed in our sample as we were able to rapidly organize home therapy. We thus decided to verify, through a questionnaire, the compliance with therapy and the degree of satisfaction of our sample of patients treated at home. From the evaluation of the answers provided to the questionnaire, an undeniably positive role of home therapy emerges as a good strategy that allowed the patients to be treated in a familial, comfortable, private environment with the help of a professional team. Home therapy was demonstrated to avoid patient stress, anxiety, and any possible conflict with other relatives who were previously forced to reschedule their lives to take the patients to the Centre. All these aspects improved the quality of life of the patients in this study. These data appear even more significant in a stressful period such as the current period of pandemic threats that magnify feelings of anger, emotion, anxiety, and uncertainty. Knowing that they were closely supported by family members and by professional figures (who are prone to listen and are competent in the field of their disease) made our patients confident to be sustained and understood as a person with a rare, chronic disease.

Moreover, the safety of these home infusions, which were entrusted to specialized teams, were attested by the absence or scarcity of adverse effects observed. Wilson et al. [14] compared the safety of home-based versus hospital/outpatient-based infusions in the United States. Their real-world data were retrospectively analysed to assess the safety of home-based laronidase ERT for MPS I. They found no evidence that the safety of laronidase ERT changed with home infusion. Indeed, no significant adverse event was registered in our sample.

Yet before the pandemic emergency, the usefulness of home therapy was stressed by other authors who reported a better compliance to ERT and a positive impact on the physical and psychologic well-being of patients with MPS II or Fabry or Gaucher disease [15,16,17,18].

Other recent studies seem to be in line with our findings. Patients suffering from metabolic diseases were interviewed by ad hoc questionnaires during the COVID-19 pandemic.

Andrade-Campos et al. [19], analysing the effects of the pandemic on Spanish patients with Gaucher disease, stressed the need to implement therapy, either orally or through home ERT, to ensure patients’ physical well-being and to lower psychological problems by avoiding any infections and major problems in cases of forced social segregation, such as during pandemics.

In Turkey, Kahraman AB et al. [20], highlighted how the fear of contracting COVD-19 caused patients with different lysosomal disorders to desist from going to hospitals to receive ERT, creating difficulties and worsening their condition.

In addition, a study by a Polish group on patients with Fabry disease [21] considered home infusion a potential therapeutic option to avoid possible feelings of anxiety, fear, and worry of a probable infection suggesting, that home therapy be maintained after the pandemic.

In Italy, patients with Fabry or Gaucher disease could also benefit from home ERT before the emergency period. This was not allowed for patients with Pompe disease or different forms of MPSs whose therapy was approved only in the hospital setting.

A clear improvement in “daily life skills” was attested to by our sample of patients with LSDs through reports of positive feelings and self-sufficiency. Moreover, all patients described an improved ability to solve problems without help, increasing their creativity and interaction with their environment. We also observed a better emotional self-control, leading to a new knowledge of self, and an increased ability to understand the feelings of relatives. Another positive effect was the patients’ improved view of themselves and their condition with positive feelings as a result of the safer and more comfortable environment. In summary, the future is now seen with hope and optimism, although resignation sometimes emerges.

## 5. Conclusions

Under the threats of the COVID-19 pandemic, we have learnt that patients with rare, chronic diseases such as LSDs are even more vulnerable and need new solutions. Telemedicine, for example, was a good alternative to monitoring patients [22], although it was not enough to treat a lysosomal disorder as ERT is a lifelong therapy administered by intravenous infusion every week or every other week. The need to be treated in a hospital setting, at a referral centre, is often stressful for the patients and their families. They often travel from far places and therefore the frequent visits and infusion calendars are time-consuming and disruptive. Our study focused on the effects of switching from hospital to home infusion under the threat of the pandemic. Although the questionnaire did not address those aspects strictly deriving from the awareness of a lower risk of contracting a COVID-19 infection, there is enough evidence that, as for all rare, chronic diseases, the opportunity to be treated at home improves quality of life, allows for a better self-esteem, and normalizes intrafamilial relationships. Well-organized home therapy provides the medical staff with confidence that the patient is compliant (testified by the overall small number of missed infusions in our sample) and follows the correct protocols for the management of any possible infusion-associated reactions. This was possible thanks to the help of the producing companies.

Our data underline that new therapeutic solutions are possible for chronic diseases and that home ERT has a paramount positive effect both for patients and their families. With the end of the emergency period, the patients will be asked to return to the hospital, with very negative expected impacts on their compliance. Authorities should redesign more integrated and patient-centred care pathways, considering the benefits also in terms of social costs.

## Figures and Tables

**Figure 1 healthcare-11-01176-f001:**
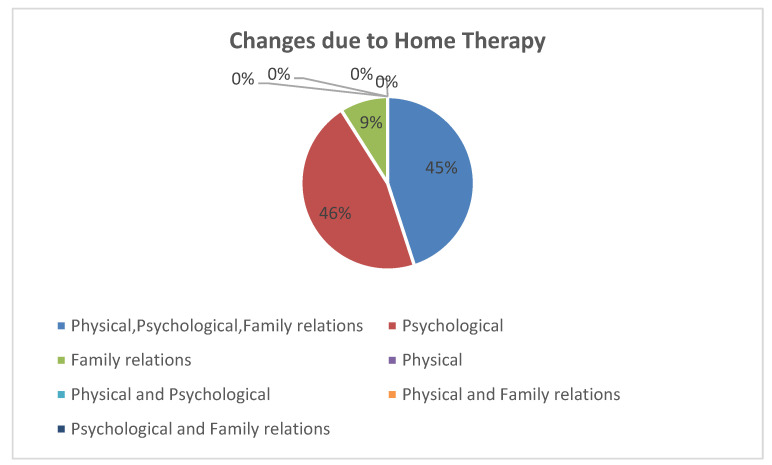
Reported changes related to home therapy in the study patients.

**Figure 2 healthcare-11-01176-f002:**
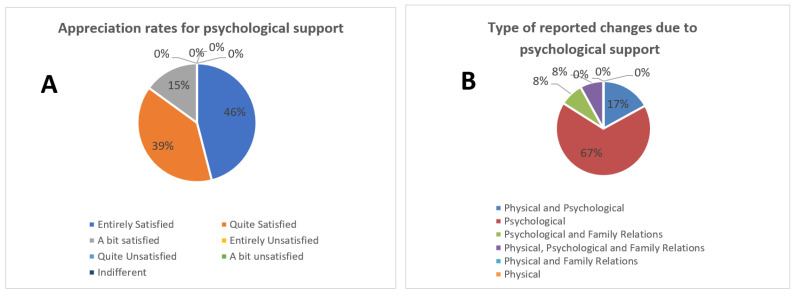
(**A**) Appreciation of psychological support. The majority of our patients were completely (46%) or quite satisfied (39%) with the psychological support and guidance offered both pre- and post-transition to their homes (**B**) Type of changes due to psychological support.

**Figure 3 healthcare-11-01176-f003:**
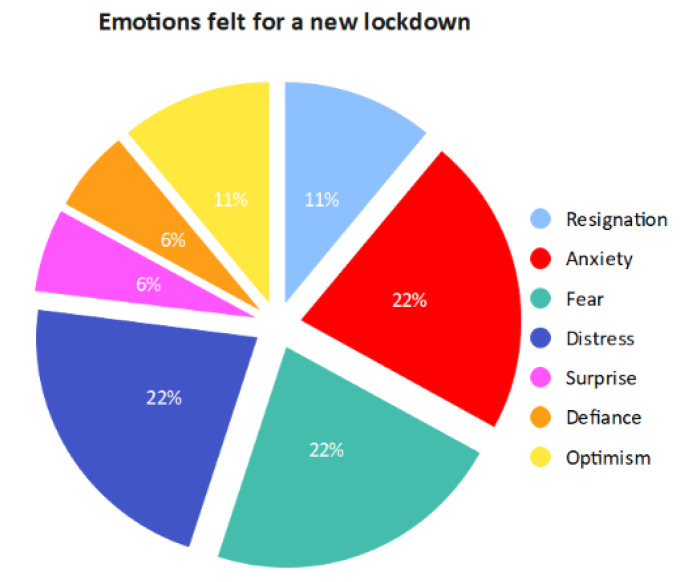
Perceived emotions related to a new lockdown risk.

## Data Availability

The raw datasets generated and/or analysed during the current study are not publicly available in order to protect participant confidentiality. For more information, please contact the authors.

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
