# Peer review of "Positive Impact of Home ERT for Mucopolysaccharidoses and Pompe Disease: The Lesson Learnt from the COVID-19 Pandemic"

_healthcare, 2023, doi:10.3390/healthcare11081176_

Round 1

Reviewer 1 Report

Did you take consent every time ERT was given
What was the back up in case if any adverse reaction occurred during ERT

Did any patient contacted Covid-19 during home ERT?

Do you recommend home ERT after the pandemic and if family wants to continue.

Author Response

Did you take consent every time ERT was given

We do ask informed consent at the beginning of every treatment and in January every year when we open a new clinical record, thus at the beginning of Methods section we added the  following sentence:

Several patients with different forms of LSD are followed at our Regional Referral Centre for Metabolic Diseases. Informed consent is regularly obtained at the start of the treatment and at the beginning of the new year, when we open a new Medical Record.

What was the back up in case if any adverse reaction occurred during ERT.

The monitoring system updates us regularly about any adverse event as well as of any change of the ERT infusion calendar to have our approval. No significant AED was registered in any of our patients

Did any patient contacted Covid-19 during home ERT?

No one of the patients was infected by  COVID19 during home ERT

To specify this, the following statement was added at the beginning of the Results section (row 108-111)

The Home Therapy staff, realized a monitoring system updating us regularly about any adverse event, as well as of any change of the ERT infusion calendar, to have our approval. No significant AED was registered in any of our patients and no one of them contracted COVID19 infection at the time of investigation

Do you recommend home ERT after the pandemic and if family wants to continue

We strongly believe that a well organized home therapy can improve the quality of life of the patients and their families assuring a good compliance to ERT: Moreover the continuous exchange of information between us and the caring team represents an important feedback both for patients and the  caring doctors

Reviewer 2 Report

The manuscript: “Positive impact of Home ERT for Mucopolysaccharidoses and Pompe disease. The lesson learnt from COVID-19 pandemia” authored by Fiumara et al is a very interesting report investigating the impact of pandemia on the monitoring of chronic diseases. Despite the tragic events the pandemics showed that new solutions are possible ant thet many patients can benefit from them.

The main question addressed by the research was how  satisfactory for the patients was switching to home enzyme infusions during relevant. The research might seem relevant to the very narrow group of patients with Mucopolysaccharidoses and Pompe disease but the results might be considered important to other chronic diseases as well. There were many reports published during the recent years on how did the Covid-19 pandemics influenced our standards of care of chronically sick patients but I am not aware of any concerning Mucopolysaccharidoses and Pompe disease. These are very rare diseases which explain the low number of participants in the study. Very important and worth appraisal is the part of the paper concerning patients’ expectations as well as  their opinions on psychological care provided by the Centre
The paper  is clearly written and supplemented by the questionnaire the Authors used to evaluate the patients’ satisfaction from the change in their treatment regime.
The conclusions  are consistent with the evidence and arguments presented. And they address the main question posed.

Author Response

Thank you very much for your appreciated words. Some aspects that you underlined were so interesting that we decided to use your sentences to stress their meaning.

Reviewer 3 Report

This is an interesting single-center descriptive study involving an ad hoc survey of self-reported patient satisfaction with switching to home ERT for lysosomal storage disorders. Therefore, the scope of the study is somewhat limited; however, I think patient viewpoints on these rare metabolic conditions are valuable to have in the literature. The authors' experience with switching to home ERT (forced by the pandemic) is also helpful.

I have the following comments:

1) The discussion emphasizes the positive effects of switching from hospital to home infusion. However, this switch happened during the pandemic and, in the introduction, the authors state that 60% of the patients had stopped coming into the hospital for regular ERT due to infection concerns, leading to profound negative emotions. This makes me think the questionnaire may not distinguish the positive effects of home infusion itself from the positive effects of regaining access to ERT and lower risk of getting COVID. Can the authors comment on this study limitation?

2) The paper notes that home ERT during the pandemic was sponsored by the drug manufacturers and I wonder if this sponsorship is going to continue indefinitely. Can the authors comment about expected impacts on the patients if they need to start coming back to the hospital for infusion?

3) The language in the article was not always clear. It needs proofreading/English language editing. The most important areas of confusion are:

Results - "lost infusions" - is this missed infusions, or doses of enzyme that were physically lost?

Results - I am not sure what "enough positive" and "fully positive" mean. Are these translations of the top two choices on a Likert scale? "Positive" and "Very positive" / "Satisfactory" and "Fully satisfactory"?

Author Response

  • The discussion emphasizes the positive effects of switching from hospital to home infusion. However, this switch happened during the pandemic and, in the introduction, the authors state that 60% of the patients had stopped coming into the hospital for regular ERT due to infection concerns, leading to profound negative emotions. This makes me think the questionnaire may not distinguish the positive effects of home infusion itself from the positive effects of regaining access to ERT and lower risk of getting COVID. Can the authors comment on this study limitation?

We wish to thank you for your valuable remarks. Our study was made following the previous one (Fium ara et al. J Clin Med, 2020) investigating the feelings of patients with these rare diseases during pandemic. In the present investigation we aimed to check those patients switched from the hospital to Home therapy to evaluate the experience of the new setting and the effects on their well being. Indeed, due to the very low number of missed infusions, we excluded that the positive feelings were related to the possibility to regain access to ERT, while the safeness of home environment was fully acknowledged and testified by the lack of cases of COVID infections in these families.

  • The paper notes that home ERT during the pandemic was sponsored by the drug manufacturers and I wonder if this sponsorship is going to continue indefinitely. Unfortunately, this will not be the case. This is why we stress the need of the National Sanitary System endorsement.

      Can the authors comment about expected impacts on the patients if they need to start coming back to the hospital for infusion? At the moment, this aspect is scarcely evaluable. As yet, a small number of patients were forced to come back to the Hospital just because one of the two companies stopped the service, and we plan to further investigate and monitor their feelings.

According to this, Conclusion was rewritten

3) The language in the article was not always clear. It needs proofreading/English language editing.

The final version of the manuscript was checked by an English native speaker